# Mean-Square Analysis with An Application to Optimal Dimension Dependence of Langevin Monte Carlo

## Abstract

Sampling algorithms based on discretizations of Stochastic Differential Equations (SDEs) compose a rich and popular subset of MCMC methods. This work provides a general framework for the non-asymptotic analysis of sampling error in 2-Wasserstein distance, which also leads to a bound of mixing time. The method applies to any consistent discretization of contractive SDEs. When applied to Langevin Monte Carlo algorithm, it establishes $\widetilde{\mathcal{O}}\left(\sqrt{d}/\epsilon\right)$ mixing time, without warm start, under the common log-smooth and log-strongly-convex conditions, plus a growth condition on the potential of target measures at infinity. This bound improves the best previously known $\widetilde{\mathcal{O}}\left(d/\epsilon\right)$ result and is optimal in both dimension $d$ and accuracy tolerance $\epsilon$ for log-smooth and log-strongly-convex target measures. Our theoretical analysis is further validated by numerical experiments.

## 1 Introduction

The problem of sampling statistical distributions has attracted considerable attention, not only in the fields of statistics and scientific computing, but also in machine learning (Robert and Casella, 2013; Andrieu et al., 2003; Liu, 2008); for example, how various sampling algorithms scale with the dimension of the target distribution is a popular recent topic in statistical deep learning (see discussions below for references). For samplers that can be viewed as discretizations of SDEs, the idea is to use an ergodic SDE whose equilibrium distribution agrees with the target distribution, and employ an appropriate numerical algorithm that discretizes (the time of) the SDE. The iterates of the numerical algorithm will approximately follow the target distribution when converged, and can be used for various downstream applications such as Bayesian inference and inverse problem (Dashti and Stuart, 2017). One notable example is the Langevin Monte Carlo algorithm (LMC), which corresponds to Euler-Maruyama discretization of overdamped Langevin equation. Its study dated back to at least the 90s (Roberts et al., 1996) but keeps on leading to important discoveries, for example, on non-asymptotics and dimension dependence, which are relevant to machine learning (e.g., Dalalyan (2017a,b); Cheng et al. (2018a); Durmus et al. (2019a,b); Vempala and Wibisono (2019); Dalalyan and Riou-Durand (2020); Erdogdu and Hosseinzadeh (2020); Mou et al. (2019)). LMC is closely related to SGD too (e.g., Mandt et al. (2017)). Many other examples exist, based on alternative SDEs and different discretizations (e.g., Dalalyan and Riou-Durand (2020); Ma et al. (2021); Mou et al. (2021); Li et al. (2020); Roberts and Rosenthal (1998); Chewi et al. (2020); Shen and Lee (2019)).

Quantitatively characterizing the non-asymptotic sampling error of numerical algorithms is usually critical for choosing the appropriate algorithm for a specific downstream application, for providing practical guidance on hyperparameter selection and experiment design, and for designing improved samplers. A powerful tool that dates back to (Jordan et al., 1998) is a paradigm of non-asymptotic error analysis, namely to view sampling as optimization in probability space, and it led to many

important recent results (e.g., Liu and Wang (2016); Dalalyan (2017a); Wibisono (2018); Zhang et al. (2018); Frogner and Poggio (2020); Chizat and Bach (2018); Chen et al. (2018); Ma et al. (2021); Erdogdu and Hosseinzadeh (2020)). It works by choosing an objective functional, typically some statistical distances/diverges, and showing that the law of the iterates of sampling algorithms converges in that objective functional. However, the choice of the objective functional often needs to be customized for different sampling algorithms. For example, KL divergence works for LMC (Cheng and Bartlett, 2018), but a carefully hand-crafted cross term needs to be added to KL divergence for analyzing KLMC (Ma et al., 2021). Even for the same underlying SDE, different discretization schemes exist and lead to different sampling algorithms, and the analyses of them had usually been case by case (e.g., Cheng et al. (2018b); Dalalyan and Riou-Durand (2020); Shen and Lee (2019)). Therefore, it would be a desirable complement to have a unified, general framework to study the non-asymptotic error of SDE-based sampling algorithms.

As an important member of the family of SDE-based sampling algorithms, Langevin Monte Carlo is widely used in practice. Its stochastic gradient version is implemented in common machine learning systems, such as Tensorflow (Abadi et al., 2016), and is the off-the-shelf algorithm for large scale Bayesian inference. With the ever-growing size of parameter space, the non-asymptotic error of LMC is of central theoretical and practical interest, in particular, its dependence on the dimension of the sample space. The best current known upper bound of the mixing time in 2-Wasserstein distance for LMC is $\widetilde{\mathcal{O}}\left(\frac{d}{\epsilon}\right)$ (Durmus et al., 2019b). Motivated by a recent result (Chewi et al., 2020) that shows better dimension dependence for a Metropolis-Adjusted improvement of LMC, we wonder if the current bound for (unadjusted) LMC is tight, and if not, what is the optimal dimension dependence?

**Our contribution**    We study a broad family of numerical algorithms that discretize SDEs that have a contraction property (possibly after a coordinate transformation). For this type of problems, we revisit the classical mean-square analysis (Milstein and Tretyakov, 2013) in numerical SDE literature and extend its the global error bound from finite time to infinite time. Same as in classical mean-square analysis, we show the global error is only half order lower than the order of local strong error ($p_2$). We further obtain a $\widetilde{\mathcal{O}}\left(C^{\frac{1}{p_2-\frac{1}{2}}}\frac{1}{\epsilon^{\frac{1}{p_2-\frac{1}{2}}}}\right)$ mixing time upper bound in 2-Wasserstein distance for the family of algorithms, where $C$ is a constant containing various information of the underlying problem, e.g., the dimension $d$.

As an application of the general mixing time result, we study the widely used Langevin Monte Carlo algorithm (LMC) for sampling from a Gibbs distribution $\mu \propto \exp\left(-f(\boldsymbol{x})\right)$, which is an Euler-Maruyama discretization of Langevin dynamics. Under the standard smoothness and strong-convexity assumptions, plus an additional linear growth condition on the third-order derivative of $f$, we obtain a $\widetilde{\mathcal{O}}\left(\frac{\sqrt{d}}{\epsilon}\right)$ mixing time in 2-Wasserstein distance, which improves upon the previously best known $\widetilde{\mathcal{O}}\left(\frac{d}{\epsilon}\right)$ result (Durmus et al., 2019b). For a comparison, note it was known that discretized **kinetic** Langevin dynamics can lead to $\sqrt{d}$ dependence on dimension (Cheng and Bartlett, 2018; Dalalyan and Riou-Durand, 2020) and some believe that it is the introduction of momentum that improves the dimension dependence, but our result shows that discretized overdamped Langevin (no momentum) can also have mixing time scaling like $\sqrt{d}$. In fact, it is important to mention that it was recently shown that Metropolis-Adjusted Euler-Maruyama discretization of **overdamped** Langevin (i.e., MALA) has an optimal dimension dependence of $\widetilde{\mathcal{O}}\left(\sqrt{d}\right)$ (Chewi et al., 2020), while what we analyze here is the **unadjusted** version (i.e., LMC), and it has the same dimension dependence (note however that our $\epsilon$ dependence is not as good as that for MALA; more discussion in Section 4). We also constructed an example that shows that the mixing time of LMC is at least $\widetilde{\Omega}\left(\frac{\sqrt{d}}{\epsilon}\right)$. Hence, our mixing time bound has the optimal dependence on both $d$ and $\epsilon$. Our theoretical analysis is further validated by empirical investigation of numerical examples.

## 2   Preliminaries

**Notation**    Use the symbol $\boldsymbol{x}$ to denote a $d$-dimensional vector, and the plain symbol $x$ to denote a scalar variable. Use $\|\boldsymbol{x}\|$ to denote the Euclidean norm of vector $\boldsymbol{x}$. A numerical algorithm is denoted

by $\mathcal{A}$ and its $k$-th iterate is denoted by $\bar{\boldsymbol{x}}_k$. We slightly abuse notation by identifying measures with their density function w.r.t. Lebesgue measure. We use the convention $\widetilde{\mathcal{O}}(\cdot) = \mathcal{O}(\cdot) \log^{\mathcal{O}(1)}(\cdot)$, i.e., the $\widetilde{\mathcal{O}}(\cdot)$ notation ignores the dependence on logarithmic factors. We use the notation $\widetilde{\Omega}(\cdot)$ similarly. Denote 2-Wasserstein distance by $W_2(\mu_1, \mu_2) = \left( \inf_{(\boldsymbol{X}, \boldsymbol{Y}) \sim \Pi(\mu_1, \mu_2)} \mathbb{E} \|\boldsymbol{X} - \boldsymbol{Y}\|^2 \right)^{\frac{1}{2}}$, where $\Pi(\mu_1, \mu_2)$ is the set of couplings, i.e. all joint measures with $X$ and $Y$ marginals being $\mu_1$ and $\mu_2$. Denote the target distribution by $\mu$ and the law of a random variable $\boldsymbol{X}$ by $\mathrm{Law}(\boldsymbol{X})$. Finally, denote the mixing time of an sampling algorithm $\mathcal{A}$ converging to its target distribution $\mu$ in 2-Wasserstein distance by $\tau_{\mathrm{mix}}(\epsilon; W_2; \mathcal{A}) = \inf\{k \geq 0 | W_2(\mathrm{Law}(\bar{\boldsymbol{x}}_k), \mu) \leq \epsilon\}$.

**SDE for Sampling**  Consider a general SDE

$$d\boldsymbol{x}_t = \boldsymbol{b}(t, \boldsymbol{x}_t)dt + \boldsymbol{\sigma}(t, \boldsymbol{x}_t)d\boldsymbol{B}_t \tag{1}$$

where $\boldsymbol{b} \in \mathbb{R}^d$ is a drift term, $\boldsymbol{\sigma} \in \mathbb{R}^{d \times l}$ is a diffusion coefficient matrix and $\boldsymbol{B}_t$ is a $l$-dimensional Wiener process. Under mild condition (Pavliotis, 2014, Theorem 3.1), there exists a unique strong solution $\boldsymbol{x}_t$ to Eq. (1). Some SDEs admit geometric ergodicity, so that their solutions converge exponentially fast to a unique invariant distribution, and examples include the classical overdamped and kinetic Langevin dynamics, but are not limited to those (e.g., Mou et al. (2021); Li et al. (2020)). Such SDE are desired for sampling purposes, because one can set the target distribution to be the invariant distribution by choosing an SDE with an appropriate potential, and then solve the solution $\boldsymbol{x}_t$ of the SDE and push the time $t$ to infinity, so that (approximate) samples of the target distribution can be obtained. Except for a few known cases, however, explicit solutions of Eq. (1) are elusive and we have to resort to numerical schemes to simulate/integrate SDE. Such example schemes include, but are not limited to Euler-Maruyama method, Milstein methods and Runge-Kutta method (e.g., Kloeden and Platen (1992); Milstein and Tretyakov (2013)). With constant stepsize $h$ and at $k$-th iteration, a typical numerical algorithm takes a previous iterate $\bar{\boldsymbol{x}}_{k-1}$ and outputs a new iterate $\bar{\boldsymbol{x}}_k$ as an approximation of the solution $\boldsymbol{x}_t$ of Eq. (1) at time $t = kh$.

**Langevin Monte Carlo Algorithm**  LMC algorithm is defined by the following update rule

$$\bar{\boldsymbol{x}}_k = \bar{\boldsymbol{x}}_{k-1} - h\nabla f(\bar{\boldsymbol{x}}_{k-1}) + \sqrt{2h}\boldsymbol{\xi}_k, \quad k = 1, 2, \cdots \tag{2}$$

where $\{\boldsymbol{\xi}_k\}_{k \in \mathbb{Z}_{>0}}$ are i.i.d. standard $d$-dimensional Gaussian vectors. LMC corresponds to an Euler-Maruyama discretization of the continuous overdamped Langevin dynamics $d\boldsymbol{x}_t = -\nabla f(\boldsymbol{x}_t)dt + \sqrt{2}d\boldsymbol{B}_t$, which converges to an equilibrium distribution $\mu \sim \exp(-f(\boldsymbol{x}))$.

Dalalyan (2017b) provided a non-asymptotic analysis of LMC. An $\widetilde{\mathcal{O}}\left(\frac{d}{\epsilon^2}\right)$ mixing time bound in $W_2$ for log-smooth and log-strongly-convex target measures (Dalalyan, 2017a; Cheng et al., 2018a; Durmus et al., 2019a) has been established. It was further improved to $\widetilde{\mathcal{O}}\left(\frac{d}{\epsilon}\right)$ under additional Lipschitz assumption on the Hessian of $f$ (Durmus et al., 2019b). Mixing time bounds of LMC in other statistical distances/divergences have also been studied, including total variation distance (Dalalyan, 2017b; Durmus et al., 2017) and KL divergence (Cheng and Bartlett, 2018).

**Classical Mean-Square Analysis**  A powerful framework for quantifying the *global* discretization error of a numerical algorithm for Eq. (1), i.e., $e_k = \left\{ \mathbb{E} \|\boldsymbol{x}_{kh} - \bar{\boldsymbol{x}}_k\| \right\}^{\frac{1}{2}}$, is mean-square analysis (e.g., the monograph of Milstein and Tretyakov (2013)). Mean-square analysis studies how *local* integration error propagate and accumulate into global integration error; in particular, if one-step (local) weak error and strong error (both the exact solution $\boldsymbol{x}_t$ and the numerical approximation start from the same initial value $\boldsymbol{x}$) satisfy

$$\begin{aligned}
\|\mathbb{E}\boldsymbol{x}_h - \mathbb{E}\bar{\boldsymbol{x}}_1\| \leq& C_1 \left(1 + \mathbb{E} \|\boldsymbol{x}\|^2\right)^{\frac{1}{2}} h^{p_1}, \quad \text{(local weak error)} \\
\left(\mathbb{E} \|\boldsymbol{x}_h - \bar{\boldsymbol{x}}_1\|^2\right)^{\frac{1}{2}} \leq& C_2 \left(1 + \mathbb{E} \|\boldsymbol{x}\|^2\right)^{\frac{1}{2}} h^{p_2}, \quad \text{(local strong error)}
\end{aligned} \tag{3}$$

over a time interval $[0, Kh]$ for some constants $C_1, C_2 > 0$, $p_2 \geq \frac{1}{2}$ and $p_1 \geq p_2 + \frac{1}{2}$, then the global error can be bounded by $e_k \leq C \left(1 + \mathbb{E} \|\boldsymbol{x}_0\|^2\right)^{\frac{1}{2}} h^{p_2 - \frac{1}{2}}$, $k = 1, 2, \cdots, K$ for some constant $C > 0$ dependent on $Kh$.

Although classical mean-square analysis is only concerned with numerical integration error, sampling error can be also inferred. However, there is a limitation that prevents directly employing mean-square analysis in the non-asymptotic analysis of sampling algorithms. The bound of global error only holds in finite time because the constant $C$ can grow exponentially as $K$ increases, rendering the bound useless when $K \to \infty$.

# 3 Mean-Square Analysis of Samplers Based on Contractive SDE

In order to prepare for the analysis of **sampling** error, we first show that the finite time limitation of **integration** error analysis can be lifted if the SDE being discretized is contractive.

More precisely, one bottleneck that prevents the results of classical mean-square analysis from extending to infinite time horizon, is the fact that the solution of a general SDE may not be bounded, and neither is its discretization. Note that local error (Eq. (3)) depends on the initial value. To go from local to global error, these 'initial' values correspond to iterates of numerical algorithms, which change from iteration to iteration and can be unbounded, hence when accumulated together, it is possible that the global error may blow up.

Samplers considered here, on the other hand, are based on stochastic differential equations, each of which weakly converges to a limiting distributions. The solution of the underlying converging SDE, as it converges to the invariant measure, gradually inherits boundedness properties from the target measure. Thus, as long as the target measure has bounded 2nd-moment, a sampling algorithm based on a reasonable discretization of the SDE should also have bounded 2nd-moment. Motivated by this observation, we will assume the sampling algorithms we study are based on contractive SDEs, which is a sufficient condition to ensure the underlying SDE converges to a statistical distribution.

**Definition 3.1.** *A stochastic differential equation is contractive if there exists a non-singular constant matrix $A \in \mathbb{R}^{d \times d}$, a constant $\beta > 0$, such that any pair of solutions of the SDE satisfy*

$$\left( \mathbb{E} \left\| A \left( \boldsymbol{x}_t - \boldsymbol{y}_t \right) \right\|^2 \right)^{\frac{1}{2}} \leq \left\| A \left( \boldsymbol{x} - \boldsymbol{y} \right) \right\| \exp(-\beta t), \tag{4}$$

*where $\boldsymbol{x}_t, \boldsymbol{y}_t$ are two solutions, driven by the same Brownian motion but evolved respectively from initial conditions $\boldsymbol{x}$ and $\boldsymbol{y}$.*

**Remark.** *As long as $\boldsymbol{b}$ and $\boldsymbol{\sigma}$ in (1) are not explicitly dependent on time, it suffices to find an arbitrarily small $t_0 > 0$ and show (4) holds for all $t < t_0$.*

**Remark.** *Sometimes contraction is not easy to establish directly, but can be shown after an appropriate coordinate transformation, see (Dalalyan and Riou-Durand, 2020, Proposition 1) for such a treatment for kinetic Langevin dynamics. The introduction of $A$ permits such transformations.*

We now use contractivity to remove the finite time limitation. We will first need a lemma, which is a local (short time) result.

**Lemma 3.2.** *(Milstein and Tretyakov, 2013, Lemma 1.3) Suppose $\boldsymbol{b}$ and $\boldsymbol{\sigma}$ in Eq.(1) are Lipschitz continuous. For two solutions $\boldsymbol{x}_t, \boldsymbol{y}_t$ of Eq. (1) starting from $\boldsymbol{x}, \boldsymbol{y}$ respectively, denote $\boldsymbol{z} := (\boldsymbol{x}_t - \boldsymbol{x}) - (\boldsymbol{y}_t - \boldsymbol{y})$, then there exist $C_0 > 0$ and $h_0 > 0$ such that*

$$\mathbb{E} \left\| \boldsymbol{z} \right\|^2 \leq C_0 \left\| \boldsymbol{x} - \boldsymbol{y} \right\|^2 t, \quad \forall \boldsymbol{x}, \boldsymbol{y}, \, 0 < t \leq h_0. \tag{5}$$

Then we will have a sequence of results that connects **sampling** error (a statistical property) with local **integration** error (a simulation property). This justifies our generic produce for non-asymptotic sampling error analysis, which only requires bounding the orders of local weak and strong integration errors (in addition to establishing contractivity of the continuous dynamics).

**Theorem 3.3.** *(**Global Integration Error, Infinite Time Version**) Suppose Eq.(1) is contractive with rate $\beta$ and with respect to a non-singular matrix $A \in \mathbb{R}^{d \times d}$, with Lipschitz continuous $\boldsymbol{b}$ and $\boldsymbol{\sigma}$, and there is a numerical algorithm $\mathcal{A}$ with step size $h$ simulating the solution $\boldsymbol{x}_t$ of the SDE, whose iterates are denoted by $\bar{\boldsymbol{x}}_k, k = 0, 1, \cdots$. Suppose there exists $0 < h_0 \leq 1, C_1, C_2 > 0, D_1, D_2 \geq 0, p_1 \geq 1, \frac{1}{2} < p_2 \leq p_1 - \frac{1}{2}$ such that for any $0 < h \leq h_0$, the algorithm $\mathcal{A}$ has, respectively, local weak and strong error of order $p_1$ and $p_2$, defined as*

$$\begin{cases} \left\| \mathbb{E} \left( \boldsymbol{x}_h - \bar{\boldsymbol{x}}_1 \right) \right\| \leq \left( C_1 + D_1 \sqrt{\mathbb{E} \left\| \boldsymbol{x} \right\|^2} \right) h^{p_1}, \\ \left( \mathbb{E} \left\| \boldsymbol{x}_h - \bar{\boldsymbol{x}}_1 \right\|^2 \right)^{\frac{1}{2}} \leq \left( C_2^2 + D_2^2 \mathbb{E} \left\| \boldsymbol{x} \right\|^2 \right)^{\frac{1}{2}} h^{p_2}, \end{cases} \tag{6}$$

173 *where $\boldsymbol{x}_h$ solves Eq.(1) with any initial value $\boldsymbol{x}$ and $\bar{\boldsymbol{x}}_1$ is the result of applying $\mathcal{A}$ to $\boldsymbol{x}$ for one step.*

174 *If the solution of SDE $\boldsymbol{x}_t$ and algorithm $\mathcal{A}$ both start from $\boldsymbol{x}_0$, then for $0 < h \leq h_1 \triangleq$*

175 $\min\left\{h_0, \frac{1}{4\beta}, \left(\frac{\sqrt{\beta}}{4\sqrt{2}\kappa_A D_2}\right)^{\frac{1}{p_2 - \frac{1}{2}}}, \left(\frac{\beta}{8\sqrt{2}\kappa_A(D_1+C_0 D_2)}\right)^{\frac{1}{p_2 - \frac{1}{2}}}\right\}$, *the global error $\boldsymbol{e}_k$ is bounded as*

176

$$e_k := \left(\mathbb{E}\|\boldsymbol{x}_{kh} - \bar{\boldsymbol{x}}_k\|^2\right)^{\frac{1}{2}} \leq Ch^{p_2 - \frac{1}{2}}, \quad k = 0, 1, 2, \cdots \tag{7}$$

177 *where*

$$C = \frac{2}{\sqrt{\beta}}\kappa_A^2 \left(\frac{C_1 + C_0 C_2 + \sqrt{2}U(D_1 + C_0 D_2)}{\sqrt{\beta}} + C_2 + \sqrt{2}D_2 U\right), \tag{8}$$

178 $C_0$ *is from Eq.* (5), $\kappa_A$ *is the condition number of matrix $A$ and $U^2 \triangleq 4\|\boldsymbol{x}_0\|^2 + 5\mathbb{E}_\mu \|\boldsymbol{x}\|^2$.*

179 **Remark.** *We use the convention $1/0 = \infty$ when $D_1 = D_2 = 0$. This is pertinent when a numerical*
180 *algorithm $\mathcal{A}$, e.g. LMC (Lemma D.3), produces bounded iterates. In such cases, the initial value in*
181 *Eq.* (6) *are iterations of $\mathcal{A}$ and will be bounded, it then can be absorbed into $C_1, C_2$ and we may set*
182 $D_1 = D_2 = 0$.

183 Following Theorem 3.3, we obtain the following non-asymptotic bound of the sampling error in $W_2$:

184 **Theorem 3.4.** *(**Non-Asymptotic Sampling Error Bound: General Case**) Under the same assump-*
185 *tion and with the same notation of Theorem 3.3, we have*

$$W_2(Law(\bar{\boldsymbol{x}}_k), \mu) \leq \sqrt{2}e^{-\beta kh}W_2(Law(\boldsymbol{x}_0), \mu) + \sqrt{2}Ch^{p_2 - \frac{1}{2}}, \quad \forall 0 < h \leq h_1.$$

186 A corollary of Theorem 3.4 is a bound on the mixing time of the sampling algorithm:

187 **Corollary 3.5.** *(**Upper Bound of Mixing Time: General Case**) Under the same assumption and*
188 *with the same notation of Theorem 3.3, we have*

$$\tau_{\mathrm{mix}}(\epsilon; W_2; \mathcal{A}) \leq \max\left\{\frac{1}{\beta h_1}, \frac{1}{\beta}\left(\frac{2C}{\epsilon}\right)^{\frac{1}{p_2 - \frac{1}{2}}}\right\} \log\frac{2\sqrt{2}W_2(Law(\boldsymbol{x}_0)\mu)}{\epsilon}$$

189 *In particular, when high accuracy is needed, i.e., $\epsilon < 2Ch_1^{p_2 - \frac{1}{2}}$, we have*

$$\tau_{\mathrm{mix}}(\epsilon; W_2; \mathcal{A}) \leq \frac{(2C)^{\frac{1}{p_2 - \frac{1}{2}}}}{\beta} \frac{1}{\epsilon^{\frac{1}{p_2 - \frac{1}{2}}}} \log\frac{2\sqrt{2}W_2(Law(\boldsymbol{x}_0), \mu)}{\epsilon} = \widetilde{\mathcal{O}}\left(\frac{C^{\frac{1}{p_2 - \frac{1}{2}}}}{\beta} \frac{1}{\epsilon^{\frac{1}{p_2 - \frac{1}{2}}}}\right) \tag{9}$$

190 Corollary 3.5 states how mixing time depends on the order of local (strong) error (i.e., $p_2$) of a
191 numerical algorithm. The larger $p_2$ is, the shorter the mixing time of the algorithm is, in term of
192 the dependence on accuracy tolerance parameter $\epsilon$. It is important to note that for constant stepsize
193 discretizations that are deterministic on the filtration of the driving Brownian motion and use only its
194 increments, there is a strong order barrier, namely $p_2 \leq 1.5$ (Rüemelin, 1982); however, methods
195 involving multiple stochastic integrals (e.g., Kloeden and Platen (1992); Milstein and Tretyakov
196 (2013)) and randomization (e.g., Shen and Lee (2019)) can yield a larger $p_2$.

197 The constant $C$ defined in Eq. (7) typically contains rich information about the underlying SDE, e.g.
198 dimension, Lipschitz constant of drift and noise diffusion, and the initial value $\boldsymbol{x}_0$ of the sampling
199 algorithm. Through $C$, we can uncover the dependence of mixing time bound on various parameters,
200 such as the dimension $d$. This will be exemplified with Langevin Monte Carlo in the next section.

## 4 Non-Asymptotic Analysis of Langevin Monte Carlo Algorithm

202 This section quantifies how LMC samples from Gibbs target distribution $\mu \sim \exp\left(-f(\boldsymbol{x})\right)$ that has
203 a finite second moment, i.e., $\int_{\mathbb{R}^d} \|\boldsymbol{x}\|^2 d\mu < \infty$. Assume without loss of generality that the origin is
204 a local minimizer of $f$, i.e. $\nabla f(\mathbf{0}) = \mathbf{0}$; this is for notational convenience in the analysis and can
205 be realized via a simple coordinate shift, and it is not needed in the practical implementation. In
206 addition, we assume the following two conditions hold:

**A 1.** *(**Smoothness and Strong Convexity**) Assume $f \in \mathcal{C}^2$ and is $L$-smooth and $m$-strongly-convex, i.e. there exists $0 < m \leq L$ such that $mI_d \preccurlyeq \nabla^2 f(\boldsymbol{x}) \preccurlyeq LI_d, \quad \forall \boldsymbol{x} \in \mathbb{R}^d$.*

Denote the condition number of $f$ by $\kappa \triangleq \frac{L}{m}$. The smoothness and strong-convexity assumption is the standard assumption in the literature of analyzing LMC algorithm (Dalalyan, 2017a,b; Cheng and Bartlett, 2018; Durmus et al., 2019a,b).

**A 2.** *(**Linear Growth of the 3rd-order derivative**) Assume $f \in \mathcal{C}^3$ and the operator $\nabla(\Delta f)$ grows at most linearly, i.e., there exists a constant $G > 0$ such that $\left\| \nabla(\Delta f(\boldsymbol{x})) \right\| \leq G\left(1 + \|\boldsymbol{x}\|\right)$.*

**Remark.** *The linear growth (at infinity) condition on $\nabla \Delta f$ is actually not as restrictive as it appears, and in some sense even weaker than some classical condition for the existence of solutions to SDE. For example, a standard condition for ensuring the existence and uniqueness of a global solution to SDE is at most a linear growth (at infinity) of the drift (Pavliotis, 2014, Theorem 3.1). If we consider monomial potentials, i.e., $f(x) = x^p, p \in \mathbb{N}_+$, then the linear growth condition on $\nabla \Delta f$ is met when $p \leq 4$, whereas the classical condition for the existence of solutions holds only when $p \leq 2$.*

To apply mean-square analysis to study LMC algorithm, we will need to ensure the underlying Langevin dynamics is contractive, which we verify in Section C and D in the appendix. In addition, we work out all required constants to determine the $C$ in Eq. 7 explicitly in the appendix. With all these necessary ingredients, we now invoke Theorem 3.4 and obtain the following result:

**Theorem 4.1.** *(**Non-Asymptotic Error Bound: LMC**) Suppose Assumption 1 and 2 hold. LMC iteration $\bar{\boldsymbol{x}}_{k+1} = \bar{\boldsymbol{x}}_k - h\nabla f(\bar{\boldsymbol{x}}_k) + \sqrt{2h}\xi_k$ satisfies*

$$W_2(Law(\bar{\boldsymbol{x}}_k), \mu) \leq \sqrt{2}e^{-mkh}W_2(Law(\boldsymbol{x}_0), \mu) + \sqrt{2}C_{LMC}h, \quad 0 < h \leq \frac{1}{4\kappa L}, k \in \mathbb{N} \quad (10)$$

*where $C_{LMC} = \frac{10(L^2+G)}{m^{\frac{3}{2}}}\sqrt{2d + m\left(\|\boldsymbol{x}_0\|^2 + 1\right)} = \mathcal{O}(\sqrt{d})$.*

Corollary 3.5 combined with the above result gives the following bound on the mixing time of LMC:

**Theorem 4.2.** *(**Upper Bound of Mixing Time: LMC**) Suppose Assumption 1 and 2 hold. If running LMC from $\boldsymbol{x}_0$, we then have*

$$\tau_{\mathrm{mix}}(\epsilon; W_2; \mathrm{LMC}) \leq \max\{4\kappa^2, \frac{2C_{LMC}}{m}\frac{1}{\epsilon}\}\log\frac{2\sqrt{2}W_2(Law(\boldsymbol{x}_0), \mu)}{\epsilon}$$

*where $C_{LMC}$ is the same in Theorem 4.1. When high accuracy is needed, i.e., $\epsilon \leq \frac{C_{LMC}}{2m\kappa^2}$, we have*

$$\tau_{\mathrm{mix}}(\epsilon; W_2; \mathrm{LMC}) \leq \frac{2C_{LMC}}{m}\frac{1}{\epsilon}\log\frac{2\sqrt{2}W_2(Law(\boldsymbol{x}_0), \mu)}{\epsilon} = \widetilde{\mathcal{O}}\left(\frac{\sqrt{d}}{\epsilon}\right).$$

The $\widetilde{\mathcal{O}}\left(\frac{\sqrt{d}}{\epsilon}\right)$ mixing time bound in 2-Wasserstein distance improves upon the previous ones (Dalalyan, 2017a; Cheng and Bartlett, 2018; Durmus et al., 2019b,a) in the dependence of $d$ and/or $\epsilon$. If further assuming $G = \mathcal{O}(L^2)$, we then have $C_{\mathrm{LMC}} = \mathcal{O}(\kappa^2\sqrt{m}\sqrt{d})$ and Thm.4.2 shows the mixing time is $\widetilde{\mathcal{O}}\left(\frac{\kappa^2}{\sqrt{m}}\frac{\sqrt{d}}{\epsilon}\right)$, which also improves the $\kappa$ dependence in some previous results (Dalalyan, 2017a; Cheng and Bartlett, 2018) in the regime $m \leq 1$. A brief comparison is summarized in Table 1.

**Optimality** In fact, the $\widetilde{\mathcal{O}}\left(\frac{\sqrt{d}}{\epsilon}\right)$ mixing time of LMC has the optimal scaling one can expect. This is in terms of the dependence on $d$ and $\epsilon$, over the class of all log-smooth and log-strongly-convex target measures. To illustrate this, consider the following Gaussian target distribution whose potential is

$$f(\boldsymbol{x}) = \frac{m}{2}\sum_{i=1}^{d} x_i^2 + \frac{L}{2}\sum_{i=d+1}^{2d} x_i^2, \quad \text{with } m = 1, L \geq 4m. \quad (11)$$

We now establish a lower bound on the mixing time of LMC algorithm for this target measure.

**Theorem 4.3.** *(**Lower Bound of Mixing Time**) Suppose we run LMC for the target measure defined in Eq. (11) from $\boldsymbol{x}_0 = \mathbf{1}_{2d}$, then for any choice of step size $h > 0$ within stability limit, we have*

$$\tau_{\mathrm{mix}}(\epsilon; W_2; \mathrm{LMC}) \geq \frac{\sqrt{d}}{8\epsilon}\log\frac{\sqrt{d}}{\epsilon} = \widetilde{\Omega}\left(\frac{\sqrt{d}}{\epsilon}\right).$$

Table 1: Comparison of mixing time results in 2-Wassertein distance of LMC with $L$-smooth and $m$-strongly-convex potential. Constant step size is used and accuracy tolerance $\epsilon$ is small enough.

| | mixing time | Additional Assumption |
|---|---|---|
| (Dalalyan, 2017a, Theorem 1) | $\widetilde{\mathcal{O}}\left(\frac{\kappa^2}{m} \cdot \frac{d}{\epsilon^2}\right)$ | N/A |
| (Cheng and Bartlett, 2018, Theorem 1) | $\widetilde{\mathcal{O}}\left(\frac{\kappa^2}{m} \cdot \frac{d}{\epsilon^2}\right)$ | N/A |
| (Durmus et al., 2019a, Corollary 10) | $\widetilde{\mathcal{O}}\left(\frac{\kappa}{m} \cdot \frac{d}{\epsilon^2}\right)$ | N/A |
| (Durmus et al., 2019b, Theorem 8) | $\widetilde{\mathcal{O}}\left(\frac{d}{\epsilon}\right)$ [1] | $\left\|\nabla^2 f(\boldsymbol{x}) - \nabla^2 f(\boldsymbol{y})\right\| \leq \widetilde{L}\left\|\boldsymbol{x} - \boldsymbol{y}\right\|$ |
| This work (Theorem 4.2) | $\widetilde{\mathcal{O}}\left(\frac{\kappa^2}{\sqrt{m}} \cdot \frac{\sqrt{d}}{\epsilon}\right)$ | Assumption 2 and $G = \mathcal{O}(L^2)$ [2] |

Combining Theorem 4.2 and 4.3, we see that mean-square analysis provides a tight bound for LMC.

However, there is one limitation of our result – Assumption 2, which is, although mild, still extra to the standard setup. Therefore, the gap between the upper bound and the lower bound of LMC algorithm over the entire family of log-smooth and log-strongly-convex target measures is not completely closed. We tend to believe that Assumption 2 may not be essential, but rather than an artifact of our proof technique. We hope to lift this restriction in future work.

**Comparison** At least two sampling algorithms are closely related to LMC. One is Kinetic Langevin Monte Carlo algorithm (KLMC), which is discretized kinetic/underdamped Langevin dynamics, and the other is Metropolis-Adjusted Langevin Algorithm (MALA) which uses the one-step update of LMC as a proposal and then accepts/rejects them with a Metropolis-Hastings algorithm.

The $\widetilde{\mathcal{O}}\left(\frac{\sqrt{d}}{\epsilon}\right)$ mixing time in 2-Wasserstein distance of KLMC has been established for log-smooth and log-strongly-convex target measures in existing literature (Cheng et al., 2018b; Dalalyan and Riou-Durand, 2020). Due to its better dimension dependence over previously best known results of LMC, KLMC is understood to be the analog of Nesterov's accelerated gradient method for sampling (Ma et al., 2021). Our findings show that LMC is able to achieve the same mixing time, albeit under an additional growth-at-infinity condition. However, this does not say anything about whether/how KLMC accelerates LMC, as the optimality of KLMC bound is not yet clear. We also note KLMC has better condition number dependence, although the $\kappa$ dependence in our bound may not be tight.

In terms of MALA, a recent work (Chewi et al., 2020) establishes a $\widetilde{\mathcal{O}}\left(\sqrt{d}\right)$ mixing time in 2-Wasserstein distance with warm start, and the dimension dependence is shown to be optimal. We see that without the Metropolis adjustment, LMC can also achieve the optimal dimension dependence as MALA. But unlike LMC, MALA only has logarithmic dependence on $\frac{1}{\epsilon}$. Under warm-start condition, is it possible/how to improve the dependence of $\frac{1}{\epsilon}$ for LMC, from polynomial to logarithmic? This question is beyond the scope of this paper but worth further investigation.

## 5  Numerical Examples

This section numerically verifies our theoretical findings for LMC in Section 4, with a particular focus on the dependence of the discretization error in Theorem 4.1 on dimension $d$ and step size $h$. To this end, we consider two target measures specified by the following two potentials:

$$f_1(\boldsymbol{x}) = \frac{1}{2}\left\|\boldsymbol{x}\right\|^2 + \log\left(\sum_{i=1}^{d} e^{x_i}\right) \quad \text{and} \quad f_2(\boldsymbol{x}) = \frac{1}{2}\left\|\boldsymbol{x}\right\|^2 - \frac{1}{2d^{\frac{1}{2}}}\sum_{i=1}^{d}\cos\left(d^{\frac{1}{4}}x_i\right). \quad (12)$$

It is not hard to see that $f_1$ is 2-smooth and 1-strongly convex, $f_2$ is $\frac{3}{2}$-smooth and 1-strongly-convex. $f_2$ is also used in (Chewi et al., 2020) to illustrate the optimal dimension dependence of MALA. Explicit expression of 2-Wasserstein distance between non-Gaussian distributions is

---

[1]The dependence on $\kappa$ is not readily available from Theorem 8 in Durmus et al. (2019b).

[2]The $G = \mathcal{O}(L^2)$ assumption is only for $\kappa, m$ dependence. Removing it does not affect $d, \epsilon$ dependence.

274 typically not available, instead, we use the Euclidean norm of the mean error as a surrogate because
275 $\left\| \mathbb{E}\bar{\boldsymbol{x}}_k - \mathbb{E}_\mu \boldsymbol{x} \right\| \leq W_2(\mathrm{Law}(\bar{\boldsymbol{x}}_k), \mu)$ due to Jensen's inequality. To obtain an accurate estimate of the
276 ground truth, we run $10^8$ independent LMC realizations using a tiny step size (h = 0.001), each till a
277 fixed, long enough time, and use the empirical average to approximate $\mathbb{E}_\mu \boldsymbol{x}$.

278 To study the dimension dependence of sampling error, we fix step size $h = 0.1$, and for each
279 $d \in \{1, 2, 5, 10, 20, 50, 100, 200, 500, 1000\}$, we simulate $10^4$ independent Markov chains using
280 LMC algorithm for 100 iterations, which is long enough for the chain to be well-mixed. The mean
281 and the standard deviation of the sampling error corresponding to the last 10 iterates are recorded.

282 To study step size dependence of sampling error, dimension is fixed to be $d = 10$. We experiment with
283 step size $h \in \{1, 2, 3, 4, 5, 6, 7, 8, 9, 10\} \times 10^{-1}$. We fix a continuous time $T = 20$, and run LMC
284 algorithm for $\lceil \frac{T}{h} \rceil$ iterations for each $h$. The procedure is repeated $10^4$ times with different random
285 seeds to obtain independent samples. When the corresponding continuous time $t = kh > 10$, we see
286 from Eq. (10) that LMC is well converged and the sampling error is saturated by the discretization
287 error. Therefore, for each $h$, we take the last $\lceil \frac{10}{h} \rceil$ iterates and record the mean and standard deviation
288 of their sampling error.

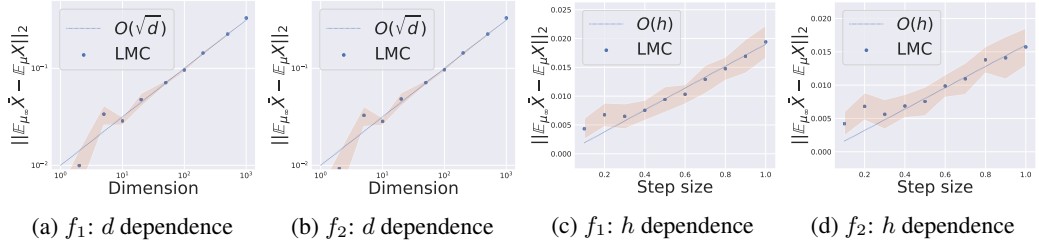

(a) $f_1$: $d$ dependence    (b) $f_2$: $d$ dependence    (c) $f_1$: $h$ dependence    (d) $f_2$: $h$ dependence

Figure 1: (a) Dependence of the sampling error of LMC on dimension $d$ and step size $h$ for $f_1$ and $f_2$.
Both axes in Figure 1a and 1b are in log scale. The shaded areas in Figure 1a and 1b represent one
standard deviation of the last 10 iterates. The shaded areas in Figure 1c and 1d represent one standard
deviation of the last $\lceil \frac{10}{h} \rceil$ iterations.

289 The experiment results shown in Figure 1 are consistent with our theoretical analysis of the sampling
290 error. Both linear dependence on $\sqrt{d}$ and $h$ can be identified in and supported by the empirical
291 evidence. Note results with smaller $h$ are less accurate because one starts to see the error of empirical
292 approximation due to finite samples. Experiments were conducted on a machine with a 2.20GHz
293 Intel(R) Xeon(R) E5-2630 v4 CPU and an Nvidia GeForce GTX 1080 GPU.

## 294    6   Conclusion

295 This paper extends the mean-square analysis framework for analyzing the integration error of SDE
296 to analyzing the sampling error in 2-Wasserstein distance. Corresponding mixing time bound
297 unveils how a high-order numerical algorithm can help improve dependence on accuracy tolerance $\epsilon$,
298 and potentially other parameters, such as the dimension. When applied to Langevin Monte Carlo
299 algorithm, it obtains an improved and optimal $\widetilde{\mathcal{O}}\left(\sqrt{d}/\epsilon\right)$ bound, which was previously thought to be
300 obtainable only with the addition of momentum.

301 Here are some possible directions worth further investigations. (i) In data-intensive applications,
302 stochastic gradients are typically used for better scalability. It seems natural to apply the mean-square
303 analysis framework to study SDE-basd stochastic gradient MCMC methods; (ii) Assumption 2 is
304 likely to be an artifact of our analysis; how to establish the optimal mixing time bound in the standard
305 log-smooth and log-strongly-convex setup is still an open question; (iii) Motivated by the recent
306 result of MALA (Chewi et al., 2020), it would be interesting to know whether the dependence on $\frac{1}{\epsilon}$
307 can be improved to logarithmic, for example if LMC is initialized at a warm start.

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
