# OpenReview forum: "Mean-Square Analysis with An Application to Optimal Dimension Dependence of Langevin Monte Carlo"
_NeurIPS.cc/2021/Conference — NeurIPS 2021 Submitted_

### Official Review · Reviewer_j2p5 · 2021-06-30

**Rating:** 4
**Confidence:** 5

**Summary:**

This paper introduces a framework for analyzing the discretization of SDEs such as the Langevin diffusion. Within this framework, once the discretization error incurred by the algorithm in a single iteration is controlled, it automatically yields a mixing time bound in Wasserstein distance for the algorithm. Importantly, this framework only applies to contractive SDEs which essentially requires the target distribution to be strongly log-concave (in particular, it cannot accommodate common weakinings of this condition, such as a log-Sobolev inequality). This framework is then applied to the standard Langevin Monte Carlo algorithm and under strong log-concavity, log-smoothness, and a condition on the third-order derivative of the potential, a mixing time bound of d^{½}/ε is obtained; this bound is shown to be tight on Gaussian targets. Prior bounds (using different third-order conditions) obtained at best d/ε.

**Limitations And Societal Impact:**

Yes.

**Main Review:**

This work is interesting; certainly the idea of reducing the problem of proving mixing time bounds to a simple computation of one step of discretization is quite appealing. However, I do not find the results in this paper entirely convincing. Without a higher-smoothness condition, the approach of this paper does not improve upon prior analyses of Langevin (in fact, the proof technique appears to be more brittle since it relies heavily on contractivity). The main new result claimed in this paper is an improved dimension dependence under third-order smoothness condition, but it is possible that the third-order smoothness parameter hides some dimension dependence (see further comments below).

Specific Comments:
- Abstract should be modified from “a growth condition on the potential of target measures” to “a growth condition on the third derivative of the potential of target measures”.
- No attempt is made to compare the third-order derivative condition used in this paper with the Hessian Lipschitz condition used in prior works. Although it may not be possible to compare these conditions in full generality, there should at least be some discussion of it. The Hessian Lipschitz condition at least has the nice property that it “tensorizes”, in the sense that if f(x) = sum_i f_i(x_i) is coordinate-wise separable, with each f_i satisfying the Hessian Lipschitz condition with constant L, then the function f does too with the same constant L (and hence it is a “dimension-independent” condition); this is not true for the third-order condition in this paper. From this it is not difficult to build examples where the constant G in this paper is much larger than the Hessian Lipschitz constant. For example, we can take the coordinate-wise separable function f where each f_i is of the form f_i(x) = (x^2 + sin x)/2, and then the Hessian Lipschitz constant is O(1) whereas the parameter G of this paper scales as d^{½}. These considerations make the claim of “optimal dimension dependence of Langevin Monte Carlo” rather unclear.
- In the vein of the above two comments, it seems premature to compare the dimension dependence of LMC with underdamped Langevin and MALA, both of which attain their dimension dependence without third-order conditions. The authors seem to think that the third-order condition is a technical artefact which can be removed, but this is simply speculation at this point (it is possible that in the future we may see a lower bound construction which shows that the O(d) mixing time of LMC using only strong log-concavity and log-smoothness is in fact tight).
- On Thm. 4.3 (lower bound on mixing time): Since m is set to 1 and L is set to 4, introducing the variables m and L makes the proof unnecessarily confusing (e.g. going from eq. (26) to (27) where all the m’s disappear). As another note, an alternative way to deduce (27) would be to use the expression for the bias of Langevin (computed in e.g. Example 2 of https://arxiv.org/abs/1802.08089) and to deduce that the step size cannot be set too large; this should hopefully simplify the calculation to the point where the lower bound can capture some dependence on the condition number (by considering a potential whose potential has d-1 eigenvalues which are m and 1 eigenvalue which is L), which would be interesting.
- The experiments are not convincing since they do not plot the Wasserstein distance but rather the difference in means, which seems like an especially poor proxy for the Wasserstein distance. Is it possible to at least plot an approximation of the Wasserstein distance, e.g. entropically-regularized Wasserstein distance?

**Time Spent Reviewing:**

1

---

> ### Author Response · Authors · 2021-08-09
> **Response to Reviwer j2p5**
>
> We deeply appreciate the reviewer's time and valuable comments.
>
> - Re: framework only applies to contractive SDEs which essentially requires the target distribution to be strongly log-concave
>
> This is a fair statement. It is indeed desirable to weaken the assumption on the target measure, e.g., requiring isoperimetric inequalities such as Poincare's inequality and log Sobolev inequality, or growth conditions on the potential, etc. However, most existing techniques for these nonconvex problems need case-by-case analyses and this is a little complementary to our goal of having a user-friendly generic framework.
>
> For example, a common way to utilize isoperimetric inequalities for analyzing non-log-concave samplers is to construct an objective functional/Lyapunov function that leverages on the special structure of the underlying stochastic process. A well-known example is the choice of KL divergence as the Lyapunov function for the geometric ergodicity of overdamped Langevin under log Sobolev assumption, but KL divergence no longer works (directly) if the dynamics is underdamped Langevin instead, and the Lyapunov function needs to be modified in a nontrivial way (e.g., Ma et al., 2021). Another powerful class of approaches, based on coupling, require the design of coupling (e.g., synchronous, reflection, maximum) and a smart partition of state space, both of which are tailored to the specific dynamics. In short, there seems to be a trade-off between the restrictiveness of the assumptions one puts on target measures, and the range of stochastic processes an approach can quantitatively analyze. As this work tries to establish a easy-to-use framework for a generic class of SDE-based sampling algorithms, we chose to work in the standard strongly convex and smooth setup. Contractivity then follows as the expert pointed out (sometimes after some additional work, such as in the case of underdamped Langevin (Dalalyan and Riou-Durand, 2020)). As there are still many open and important problems in the area, the convexity assumption, albeit strong, is still widely used in literature, e.g., LMC (Dalalyan 2017; Cheng and Bartlett 2018; Durmus and Moulines 2019a & 2019b), Underdamped Langevin (Cheng et al. 2018, Dalalyan and Riou-Durand, 2020), MALA (Dwivedi et al. 2019, Chewi et al. 2020) and Hamiltonian Monte Carlo (Mangoubi and Smith 2017, Chen and Vempala, 2019).
>
> Meanwhile, we believe there is at least one way to extend the framework and replace contractivity by a weaker requirement. Progress is along the way, but we're hoping the reviewer could kindly allow us to take step by step.
>
> - Re: no comparison between our 3rd-order derivative condition and the Hessian Lipschitz condition in the literature
>
> This helpful comment is greatly appreciated. We believe discussing these two assumptions will significantly strengthen this work and it will be included in a revision. In short, neither is always a stronger requirement than the other. More precisely, there are two levels of discussions, first being on the satisfaction of the condition, and second being on the detail of the condition, such as the constant.
>
> For the former, we note:
> 1. A2 is no stronger than Hessian Lipschitz condition, if not weaker. Consider for example $f(x)=x^4$, and $f$ satisfies A2 but is not Hessian Lipschitz.
> 2. Hessian Lipschitz + Smoothness --> A2.
>
> For the latter (detail of the constant), we fully agree that it matters a lot whether the constant $G$ in our condition and/or the Hessian Lipschitz constant (denoted by $\tilde{L}$) are implicitly dimension dependent. The great comment on the 'tensorized' special cases is very much appreciated. In fact, there are two situations, namely (1) what the review suggested, namely $f(x)=\sum_i f_i(x_i)$, i.e., independent but not identically distributed, and (2) the i.i.d. case, in which $f(x)=\sum_i f(x_i)$.
>
> For (1), the Hessian Lipschitz condition is actually not always dimension-free and does not always have better dimension-(in)dependence than our condition. An example is the following strongly-convex and smooth potential function $f(\boldsymbol{x}) = \sum_{i=1}^d \left( \frac{i^2 + 1}{2} x_i^2 + i^4 \cos (\frac{x_i}{i}) \right)$, for which the Hessian Lipschitz constant $\tilde{L}$ scales as $\mathcal{O}(d)$ whereas the constant $G$ in A2 is $G = \mathcal{O}(1)$.
>
> For (2), however, $\tilde{L}$ will be dimension free but $G$ may not be. Nonetheless, because all the dimensions decouple, our claimed $\tilde{\mathcal{O}}(\sqrt{d})$ results can be proved in a much easier way in this special case, which is why we wondered whether we really needed assumption A2 if we were smarter. But we fully agree with the review that this is just a spectulation at this point and will make sure this is clearly stated. One reason we asked ourselves whether LMC (overdamped Langevin) really has a different dimension dependence from KLMC (kinetic Langevin) is: as one lets $\gamma\to\infty$ in kinetic Langevin, the process provably converges (weakly) to overdamped Langevin (after time dilation), but dimension doesn't explicitly appear in the proof of weak convergence, and no assumption like A2 is needed either. These are of course even more spectulative but hopefully they explain why we thought maybe A2 can be weakened one day without bringing the result back to $\tilde{\mathcal{O}}(d)$.
>
> Despite of (1) or the more complicated correlated cases, the Hessian Lipschitz condition still seems to be popular in the contemporary literature (e.g., Durmus & Moulines, 2019b; Ma et al., 2021), where it is often implicitly assumed to be dimension-free. Therefore, we hope we could also assume our alternative constant $G$ to be dimension-free for now as well. Of course as the review pointed out, we need to make sure this is clarified upfront, so that results can be precisely stated without creating confusion or overclaim, and this will be done in a revision. Meanwhile, we feel that the dimension dependence issue is universal to any assumption that brings in new constants (e.g. smoothmess, strong convexity, etc.), not unique to assumption A2. We will take the reviewer's suggestion and further investigate $G$ and/or A2.
>
> - Re: experiments used a poor proxy for the Wasserstein distance.
>
> We agree that 'difference in means' is a coarse proxy; however, it has two outstanding advantages -- being computationally cheap, and being a lower bound of $W_2$ and hence also obeying the error bound in Eq. (10). Entropy-regularized $W_2$ is a great approximation of $W_2$. Unfortunately, despite its reduced cost, it still needs $\tilde{\mathcal{O}}(\frac{n^2}{\epsilon^4})$ runtime (Altschuler et al, 2017), where $n$ is the number of samples, and $\epsilon$ is the tolerance parameter. Since our experiments were designed to validate our statistical bound, the number of samples can be as large as $n=10^8$, $\epsilon$ needs to be of order $\epsilon = 10^{-2}$, and we need to record the distance in every iteration for each one of the $10^4$ independent Markov chains. Therefore, entropy-regularized Wasserstein distance is computationally infeasible to us given the scale of our experiments.
>
>
> - Re: modify the abstract.
>
> We will revise it accordingly to make it clearer. Thanks.
>
> - Re: Thm 4.3
>
> In our paper, $m$ is set to 1, yet $L$ is free as long as it is no less than 4. We are sorry for the confusion and will update the notation to make it clearer. Also, great thanks to the alternative way to deduce Eq. (27).

---

> > ### Comment · Reviewer_j2p5 · 2021-08-10
> > **Response**
> >
> > Thank you for your response. Upon further consideration, I have raised my score to a 6 because I do think the identification of a third-order condition which allows for d^{1/2} dimension dependence is interesting (and the general framework is as well). My main concern is, as stated above, that the dimension dependence is hiding in the third-order condition; however, you have assured me that this will be made clear in the final version and I am content with this.
> >
> > Out of personal interest, I would like to discuss the third-order condition further. I think your example is interesting, but also brittle; as soon as one changes the "cosine" to "sine", then the constant G is no longer dimension-independent. (By my quick calculation, when checking A2 at the origin, then this modification causes G to be of size d^{3/2}, which is worse than the Hessian Lipschitz condition.)
> >
> > While I agree that tracking the implicit dimension dependencies involved in all the parameters is neither feasible nor well-defined, I do not agree that newly introduced assumptions should be treated as dimension-free without scrutiny. There are two possible routes one can take in this situation: (1) restrict oneself to 'canonical' well-established classes, or (2) establish a matching lower complexity bound so that, sidestepping the issue of whether or not dimension dependence lurks in the assumptions, one can at least claim a sharp result for a specific class of functions. I think the sampling community is very far from obtaining anything like (2); as for (1), I don't think a canonical definition of a third-order smoothness condition for sampling has yet emerged, although the Hessian Lipschitz condition is the closest contender. Failing (1) and (2), the best option is to discuss these issues carefully upfront.
> >
> > With these considerations in mind, it's not clear that d^{1/2} result, as it stands, applies to anything more than just Gaussians and a handful of artificial examples. To be clear, I still think the result is worthwhile. However, it means that claims of optimality should not be thrown around too lightly.
> >
> > Your thoughts on LMC and KLMC are interesting. However, I'd like to point out that the larger γ is, the smaller we must take the step size in KLMC in order for the discretization to be stable (see e.g. https://arxiv.org/pdf/1807.09382.pdf). It is possible that KLMC takes advantage of finite γ in order to take much larger steps, and thus enjoy better dimension dependence. Therefore, I don't think your informal argument for LMC vs. KLMC is convincing. Of course, I don't have a convincing argument why the dimension dependence of LMC (under just second-order assumptions) should be d, so this is just speculation on my part as well.

---

> > > ### Author Response · Authors · 2021-08-10
> > > **Additional Response**
> > >
> > > Thanks very much for the additional discussions. We are glad to hear that the identification of a third-order condition which allows for $d^{1/2}$ dimension dependence is interesting (and the general framework is as well). Your reconsideration is deeply appreciated.
> > >
> > > We agree that the example is special. We are still thinking about ways to soften A2, if not to remove it.
> > >
> > > The comments on two possible routes of dealing with implicit dimension dependencies in parameters are valuable and insightful. We definitely will state the assumption and its possible ramification clearly upfront. This is actually deeply appreciated as we're suggesting a significant result and we also would like to be precise but not overclaiming.
> > >
> > > Regarding the discussion on overdamped and kinetic Langevin, we fully agree that discretization needs to be considered and it is actually the core of the problem. Every word the reviewer said is true, and a lot of complications can arise. Still, we keep on asking ourselves how can dimension enter the discretization; if the step size is chosen to be the traditional $o(\frac{1}{L})$ value, perhaps the dimension actually does not appear, unless it is lurking inside $L$? In any case, this paragraph is purely for brainstorming purposes; it is not science yet but just speculative.

---

### Official Review · Reviewer_8SkQ · 2021-07-13

**Rating:** 5
**Confidence:** 5

**Summary:**

This paper introduced a mean-square analysis to study the converge rate of overdamped Langevin dynamics to the Gibbs stationary distribution. In particular, the authors proved that the convergence in $W_2$ norm is of order $\sqrt{d}/\varepsilon$ which is also optimal. This is the same order as the underdamped Langevin equation, and beats all the previous rates.

**Limitations And Societal Impact:**

The authors have adequately addressed the limitations and potential negative societal impact of their work

**Main Review:**

This paper deals with convergence rate for a class of stochastic differential equations -- contractive SDE (based on coupling). This technique allows for analyzing the discretization error at arbitrarily large times. As an application, the authors showed that the overdamped Langevin equation converges in $W_2$ norm at rate $\sqrt{d}/\varepsilon$, and this rate cannot be improved. This beats all the previously established rate for overdamped Langevin sampling, which is the same as the status of the art underdamped sampling rate. This result is interesting, novel and important with also applications to optimization (e.g. simulated annealing).

%%%%%%%%%%%%%%%%%%%%%%%%%%%%%%%%%%%%%%%%%%%%%%

After AC's comment on a previous work "Stochastic Runge-Kutta Accelerates Langevin Monte Carlo and Beyond." Advances in Neural Information Processing Systems 32 (2019): 7748-7760, I think the results of the paper is less strong than what I initially judge. I would change my score from 7 to 5.

**Time Spent Reviewing:**

45 min

---

> ### Author Response · Authors · 2021-08-09
> **Response to Reviewer 8SkQ**
>
> We deeply appreciate the reviewer's time and valuable comments. We are glad to hear the acknowledgement that this work is interesting, novel and important.
>
> Potential applications outside the sampling regime, such as  optimization, sound very interesting, and we are thankful for the suggestion.

---

### Official Review · Reviewer_LgVW · 2021-07-15

**Rating:** 5
**Confidence:** 4

**Summary:**

This paper derives an improved 2-Wasserstein convergence bound for the Langevin algorithm (ULA). As noted in the paper, the folk wisdom is that the previously known error bound for the ULA is O(d/epsilon), compared to the underdamped MCMC O(sqrt{d}/epsilon). The paper claims to improve the ULA bound in W-2 distance, matching the case of the underdamped bound. See my detailed comments for questions.

**Limitations And Societal Impact:**

No societal impact.

**Main Review:**

1) Looking at Theorem 4.1, the main result for the LMC, the authors establish an O(\sqrt{d}) error bound. They claim that this bound improves the best known O(d) bound established in several references. However, looking at Theorem 8 of Durmus et al (2019b) (as referenced in Table 1), the error bound is derived in that paper is in terms of **squared** Wasserstein-2 (W_2^2) rather than Wasserstein 2 as in Theorem 4.1 of this paper. Taking square roots in Theorem 8 of Durmus et al (2019b) gives a O(sqrt{d}) bound. This is also confusing regarding the claims that underdamped MCMC is better than ULA. I would really appreciate the clarification regarding this point and why the bound derived in this paper is better than Theorem 8 of Durmus et al (2019b) once a square root is taken. Since this is the main claim of the paper, if this is incorrect, then the whole story needs to be rewritten. My final evaluation will mostly depend on this and I am open to increase it given a clear rebuttal.

2) An obvious question is the verifiability of A2, i.e., for the examples considered, can you verify A2 and prove that it holds for simple scenarios, e.g., the one considered in Sec. 5?

**Time Spent Reviewing:**

6

---

> ### Author Response · Authors · 2021-08-09
> **Response to Reviewer LgVW**
>
> We deeply appreciate the reviewer's time and valuable comments.
>
> - Re: Theorem 8 of Durmus and Moulines (2019b) already gave an $\tilde{\mathcal{O}}(\sqrt{d})$ bound.
>
> We are deeply thankful for the opportunity of clarification. The result (Theorem 8) in Durmus and Moulines' milestone paper actually says that the mixing time of LMC is $\tilde{\mathcal{O}}(d)$ in 2-Wasserstein distance, NOT in 2-Wasserstein squared. Table 1 of **their** paper (not ours) summarized this.
>
> For more details, note that the right hand side of Theorem 8 in Durmus and Moulines (2019b) consists of two terms, the first term is $\mathcal{O}(d)$, yet the second term (i.e. $u_n^{(3)}(\gamma)$) is $\mathcal{O}(d^2)$ (there is a leading $d$ in Eq. (11), and also another factor of $d$ near the end of the first line of Eq. (11); typeset here refers to their final arXiv version). Therefore, the right hand side of Theorem 8 is actually $\mathcal{O}(d^2)$, and after taking the square root, one ends up with $\mathcal{O}(d)$ mixing time in $W_2$. It may be easier to see this fact from Corollary 9 in Durmus and Moulines (2019b), which serves as a constant step size version of Theorem 8 and is more comparable to Theorem 4.1 of our paper (which also uses constant step size).
>
> Once this confusion is cleared up, we see that prior to this paper, the best known mixing time bound of LMC in $W_2$ is $\tilde{\mathcal{O}}(d)$, whereas a discretization of ULD (Cheng et al. 2018b; Dalalyan and Riou-Durand, 2020) can achieve $\tilde{\mathcal{O}}(\sqrt{d})$ mixing time bound in $W_2$. Hence we have been thinking whether ULD algorithm truly has better dimension dependence than LMC algorithm.
>
> It seems that the interpretation of Theorem 8 in Durmus and Moulines (2019b) is a major concern in the review and may have negatively affected the rating. We hope the above clarification could turn the doubts away, and we would deeply appreciate it if the reviewer could kindly re-evaluate the contribution of our work.
>
> - Re: Verifiability of A2 (3rd order derivative condition)
>
> This is a great suggestion. It helps us show that A2 is in fact not a very strong assumption: after direct (although lengthy) calculations, we found that potentials considered in both examples in Sec. 5 satisfy A2. We will revise accordingly, and the comment is very much appreciated.

---

> > ### Comment · Reviewer_LgVW · 2021-08-10
> > **Theorem 5**
> >
> > Thank you for your reply. It is true that your bound is better than Theorem 8 of Durmus and Moulines. In my review, I confused Theorem 8 with Theorem 5 of that paper. It looks like Theorem 5 (not Theorem 8) of Durmus and Moulines (2019b) gives a O(sqrt{d}) -- and derived under weaker assumptions than Theorem 8 of the same paper. I just wonder how this bound compares to your bound, as it looks like already O(sqrt{d}) -- again I might have missed something and would be happy to be corrected.
> >
> > Thanks.

---

> > > ### Author Response · Authors · 2021-08-10
> > > **Response to Theorem 5**
> > >
> > > Thanks very much for the additional discussions. We are glad that the confusion with Theorem 8 in Durmus and Moulines 2019b has cleared up.
> > >
> > > Regarding Theorem 5 in their paper, it leads to an $\tilde{\mathcal{O}}(\frac{d}{\epsilon^2})$ mixing time bound, which is worse than that produced by Theorem 8 which required an additional regularity assumption. This is summarized in Table 1 of **their** paper (not ours).
> > >
> > > To explain the details, we agree that the right hand side of the main result of Theorem 5 is $\mathcal{O}(d)$, and after square rooting one gets an $\mathcal{O}(\sqrt{d})$ bound of $W_2$. However, this bound on $W_2$ does **not** directly translate into the final dimension dependence of *mixing time*, because one needs to further solve for a lower bound of the number of iterations needed for the algorithm to reach $\epsilon$-neighborhood of the target measure in $W_2$ topology.
> > >
> > > More precisely, let us investigate the non-asymptotic bound
> > > $$W_2(\pi_n, \pi) \leq (1 - mh)^{n} W_2(\pi_0, \pi) + C \sqrt{d} h^p$$
> > > where $C$ is a constant independent of dimension $d$ and step size $h$. The limiting sampling error, i.e., $W_2(\pi_\infty,\pi)$, will be $\mathcal{O}(\sqrt{d})$ as discussed above. But this is not enough for an $\sqrt{d}$ dependence in the *mixing time* bound. In fact, when $p=\frac{1}{2}$, one obtains $\tilde{\mathcal{O}}(\frac{d}{\epsilon^2})$ mixing time in $W_2$, which is Durmus & Moulines 2019b Theorem 5's result (see also e.g., Theorem 1 in Dylalyan 2017a). However, if it can be shown that $p=1$, one will obtain $\tilde{\mathcal{O}}(\frac{\sqrt{d}}{\epsilon})$ mixing time in $W_2$ (not $W_2^2$), and that is Theorem 4.1 in this work.

---

> > > > ### Comment · Reviewer_LgVW · 2021-08-22
> > > > **thanks**
> > > >
> > > > Thanks for the clarification. I have no further questions and I'll update my score.

---

### Official Review · Reviewer_DVEg · 2021-07-16

**Rating:** 5
**Confidence:** 4

**Summary:**

This paper first gives an analysis framework for controlling 2-Wasserstein distance of sampling algorithms coming from discretizing a continuous Langevin dynamics.
This is achieved by first considering the bias (local weak error) and mse (local strong error) of single step update.
The multi-step error accumulate exponentially, but together with the contraction guarantee, the maximum sampling error after arbitrary steps could be controlled.
Next, this paper introduces an assumption on the linear growth of the 3rd-order derivative which is shown to improve the convergence rate in terms of dependency of dimension.
The convergence analysis is carried out based on the general analysis framework.

**Limitations And Societal Impact:**

It is recommended to amend the discussion around line 249 or introduction to consider various relevant works on the dependency of dimension with different assumptions. The randomized midpoint method is of particular relevant because it requires weaker assumptions but achieves lower dependency of dimension in convergence rate (and same dependency of dimension in 2-Wasserstein error).

For LMC, an information-based argument implies that at least $O(\sqrt{d})$ dependency of dimension in 2-Wasserstein error is optimal, therefore the dependency in this paper cannot be improved by introducing new assumptions. However, it is still not clear whether the new assumption is necessary for achieving acceleration. More specifically, we haven't rule out the existence of one potential that satisfies standard smooth and strongly convex condition but requires higher order dependency of dimension. Therefore, it is better to clarify in what sense the analysis and algorithm is optimal or use the word "optimal" sparingly.

POST REBUTTAL: I thank the authors for the response. The theoretical result is novel and I think the possible extension to the randomized midpoint method makes the analysis framework even more interesting. I will raise my score from 6 to 7.

**Main Review:**

This paper is mostly well written and easy to follow.

The analysis framework is general and easy to use but the proof is not novel.
Many researchers working  on the problem of sampling have already had this idea and internalized this method in their numerical analysis of special sampling algorithms.
However, given that none of them have managed to convert it into a general framework, there is indeed novelty in this framework.

The improvement of dependency of dimension relies on the assumption on linear growth of the 3rd-order derivative.
Has similar assumption been introduced in sampling or optimization algorithm analysis before? If the author could comment on this, it may be helpful in appreciating the novelty here.

**Time Spent Reviewing:**

1 day

---

> ### Author Response · Authors · 2021-08-09
> **Respsonse to Reviewer DVEg**
>
> We deeply appreciate the reviewer's time and valuable comments. Glad to hear the acknowledgement of the generality of our proposed framework and its ease to use.
>
> - Re: have assumptions similiar to our assumption on the 3rd-order derivative of the potential been introduced before?
>
> The exact form of this assumption is new. However, it is related to, for example, the Hessian Lipschitz condition, i.e., $\|\nabla^2 f(\boldsymbol{x}) - \nabla^2f(\boldsymbol{y})\| \leq \tilde{L} \|\boldsymbol{x} - \boldsymbol{y}\|$, used in, e.g., the classical works of (Durmus et al., 2019b) and (Ma et al., 2021). Hessian Lipschitz condition + the standard smoothness assumption -> our assumption (i.e., bounded $\|\nabla (\Delta f)(\boldsymbol{x}) \|$).
>
> We appreciate this question as it indeed helps clarify our novelty and contribution. We will revise accordingly.
>
> - Re: randomized midpoint.
>
> Randomized midpoint is a highly creative discretization of underdamped Langevin and its resulting $\mathcal{O}(d^\frac{1}{3})$ dimension dependence is remarkable. Although randomized midpoint was cited in the original submission, we indeed should add more discussions around line249; will amend. Due to page limit, we chose to present our general framework (mean sq. analysis) and then used LMC discretization of overdamped Langevin as an example application; however, calculation based on our framework seemed to give $\mathcal{O}(d^\frac{1}{3})$ dependence for randomized midpoint as well (due to its 2nd-order strong error), which means our framework is indeed working.
>
> - Re: clarify in what sense the analysis and algorithm is optimal or use the word "optimal" sparingly.
>
> We fully agree. The current results have not given a tight dimension dependence of LMC for the class of smooth and strongly convex potentials, but only for a subclass that also satisfy the 3rd derivative condition. This limitation was stated in the original submission, e.g., line 244 - 248, but we will make it more visible. Fortunately, this extra assumption is not making our subclass much smaller than those considered by the frontier of the literature (kindly see reply to the 1st point above; see also Remark in line 214-219). More precisely, prior to this work, the best known mixing time result for LMC is $\tilde{\mathcal{O}}(\frac{d}{\epsilon^2})$ in W2 distance without any extra assumption, and $\tilde{\mathcal{O}}(\frac{d}{\epsilon})$ with extra assumption such as Hessian Lipschitz, as summarized in Table 1. Our result improves the mixing time bound to $\tilde{\mathcal{O}}(\frac{\sqrt{d}}{\epsilon})$ after replacing the Hessian Lipschitz assumption by a similar one. We thank the reviewer as this comment prompts us to clarify more in a revision, and hopefully now we explained our novelty, contribution, and position in the literature a little better.

---

> > ### Comment · Reviewer_DVEg · 2021-08-14
> > **Response to Authors**
> >
> > I thank the authors for the detailed response which resolves most of my concerns.
> >
> > I agree with the authors on the claim "calculation based on our framework seemed to give $O(d^{1/3})$ dependence for randomized midpoint as well". I believe this strengthens the generality of their analysis framework.
> >
> > However, does the author have any comment on the dependence of the condition number $\kappa$?
> > It seems the framework gives $W(Law(\bar{x}_k),\mu)\leq \sqrt{2}e^{-m k h/\gamma}W(Law(\bar{x}_0),\mu)+O(L^2 h^{3/2} /m)$. This induces a worse condition number dependence than [Lee et al. 2019].
> >
> > [Lee et al. 2019] Shen, Ruoqi, and Yin Tat Lee. "The randomized midpoint method for log-concave sampling." Advances in Neural Information Processing Systems 32 (2019).

---

> > > ### Author Response · Authors · 2021-08-14
> > > **Additional Response to Reviewer DVEg**
> > >
> > > Thank you very much for the reply and additional discussion. We are glad to hear that our earlier responses addressed most of the concerns and strengthened the generality of our analysis framework.
> > >
> > > The condition number dependence of randomized midpoint is a very interesting additional question. Despite that our submission chose LMC and its dimension dependence as its focused application, we thank the reviewer for this great scientific problem. We need more time to do the full analysis but  believe it is actually nontrivial, especially if one wants a tight condition number dependence. At least for LMC which is our focus application, the condition number dependence has to be tracked very carefully throughout each lemma (on two different types of local error, etc.) and then merged, and in the end we obtained a condition number dependence that is not always worse nor better than the existing results (it is just a different kind of dependence; see Table 1 for more details). Therefore, without detailed calculations for randomized midpoint, we apologize that we cannot confirm the reviewer's suggested bound at this moment, nor do we have any a priori belief that our framework will produce a dependence comparable or better to the remarkable analysis (and results) of [Shen and Lee, 2019].
> > >
> > > Nevertheless, here is some spectulation just for scientific discussion purposes: we feel that it might be possible to customize our current framework to leverage the special properties of randomized midpoint to obtain a similar result; for example, since the local order of bias of randomized midpoint is $p_1=4$, and the local order of standard deviation is $p_2=2$, the additional gap between $p_1$ and $p_2$, i.e. $p_1 - p_2 - \frac{1}{2} = \frac{3}{2}$ can be used to remove the dependence on $C_1, D_1$ in Eq.8 of our submission.
> > >
> > > In all cases, we hope the expert reviewer could allow us to focus on the general framework + application to LMC in this short conference submission. The suggestion however helps develop future research and is deeply appreciated.

---

> ### Comment · Reviewer_DVEg · 2021-09-14
> **Updated response**
>
> Due to recent discussions, I have decided to change the review for this article. I believe this article has two contributions:
> A better dimension dependence under an additional assumption, which serves as an interesting complement to existing LMC literature.
> A general framework as a tool for analyzing the global error with single step discretization error.
> As AC points out, the framework of this article shares many overlaps with [Li et. al., 2019].
> Although I agree with the authors that their framework considers non-uniform local errors and is more suitable for deriving their main conclusion, this improvement of the framework itself seems marginal. Moreover, given that this is not the first time such a framework has been proposed, I believe the analysis framework should serve as a tool rather than a main contribution for this article.
> Therefore, I recommend the authors reorganize the contribution and submit it to another venue.

---

### Comment · Area_Chair_PGTE · 2021-09-10
**Urgent response needed from authors**

Dear authors,

I would like to thank you for answering the committee's questions thus far.

I wanted to point you to an important missed reference [Li et. al., 2019], and specifically their Theorem 1, which provides an analysis based on [Milstein and Tretyakov, 2013] for a discretization of a fast decaying Ito diffusion.

Comparing [Thm 1, Li et. al., 2019] and Thms 3.3, 3.4 and Cor 3.5 in the current paper,
one notices that the results are almost identical, with very minor differences.
I acknowledge that this is not authors' only contribution and in Section 4, authors make an additional growth assumption on the third derivative and obtain a better and optimal dimension dependency for the LMC. However, their main contributions rely on the results in Section 3, specifically Thms 3.3, 3.4, and given [Thm 1, Li et. al., 2019], it decreases the novelty of this paper significantly.

I think this paper makes nice contributions, but I also think adapting the paper to this missed reference from Neurips 2019 would require a major revision of their Section 3, and invalidates some of the novelty claims made in the paper. However, I would like the authors to please respond urgently to this issue raised by the committee, so that their response can be taken into account when finalizing the decision.

Best,
AC

Xuechen Li, Yi Wu, Lester Mackey, and Murat A. Erdogdu. "Stochastic Runge-Kutta Accelerates Langevin Monte Carlo and Beyond." Advances in Neural Information Processing Systems 32 (2019): 7748-7760.

---

> ### Author Response · Authors · 2021-09-13
> **Response to urgent comment from AC**
>
> (Sorry it took us a weekend to reply as we carefully went through the important paper of [Li et al., 2019])
>
> We sincerely thank the AC for this reference. It is really appreciated because otherwise we would have missed this very important reference, which is unacceptable to ourselves, damaging our valued scholarship, and unfair to the authors of that great work.
>
> As the AC pointed out, we thought we had two contributions in the initial submission, one being the mean-square analysis framework, and the other being improving the known dimension dependence of LMC from $d$ to $\sqrt{d}$ under an additional assumption. The second remains valid and we still think it is an interesting result, but of course what we’re discussing now is the first:
>
> Foremost of all, we completely agree that [Thm 1, Li et al., 2019] and our section 3 share significant resemblance. As both are based on [Milstein and Tretyakov, 2013] (numerical analysis of SDEs), both their goal (transferring local integration error to global sampling error) and the proof techniques are similar, and of course we’re the one who needs to revise.
>
> Meanwhile, two observations pertinent to our work are worth mentioning: (i) our requirement on the local integration error is weaker; (ii) since our focus, namely the dimension dependence of LMC, is not necessarily identical to that of [Li et al. 2019], our Theorem 3.3 is actually more suitable than [Thm 1, Li et al] for tracking dimension dependence.
>
> More precisely (for (i)), note our constant $C_1$ and $C_2$ correspond to their $\lambda_1$ and $\lambda_2$ via $C_1=\sqrt{\lambda_2}$ and $C_2=\sqrt{\lambda_1}$, but we also have additional $D_1$ and $D_2$ terms which weaken the needed conditions on local accuracy (they need $D_1=D_2=0$, i.e., uniform local errors, while we can work with non-uniform local errors).
> In general, local error tends to depend on initial values, i.e. $D_1 \neq 0, D_2 \neq 0$. Consider the simplest example of 1D standard Gaussian being the target distribution, then we have
>
> $E| E[x_{k+1} - \\bar{x}_{k+1}|\\mathcal{F}_k]|$
>
> $= (e^{-h} - 1 + h) E|\\bar{x}_k | = (h^2/2 + o(h^2)) E|\\bar{x}_k| $
>
> $<= (h^2/2 + o(h^2)) \\sqrt{E|\\bar{x}_{k}|^2}$,
>
> where $x_{k+1}$ and $\\bar{x}_{k+1}$ are respectively the exact solution of Langevin dynamics and the one-step-iterate of LMC from initial value $\bar{x}_k$. One can see that the local error does depend on $\bar{x}_k$ and is not uniform, at least not directly. Our weaker local error condition (i.e. $D_1, D_2 \neq 0$) reflects this, and the proof of the final sampling error bound consequently used significantly more work.
>
> For (ii), note that even the extra dependence on $D_1$ and $D_2$ aside, the specific forms of our bounds in Theorem 3.3 do have different expressions from those in [Thm 1, Li et al]. In fact, bounds and constants in [Thm 1, Li et al] were probably not designed for tightly tracking the dimension dependence, as the focus of that great paper was more on $\epsilon$ dependence; consequently, [Thm 1, Li et al] only led to an $\tilde{O}(d)$ bound of LMC (see Example 1 of their paper), whereas our Theorem 3.3 led to $\tilde{O}(\sqrt{d})$. Working out and tracking all the dimension dependence is the hardest part of our work, and for that (see Sec.4) we did rely on the specific form of our bound in Sec.3.
>
> Having said all these, we would completely agree that our Section 3 is not worth a publication if it was not preparational for Section 4 (LMC dimension dependence). The very important paper of [Li et al] already made the first contribution despite of our technical differences, and we pledge that what’s old and new will be fairly clarified in our revision.  Now we think our contribution is mostly the second.
>
> As the AC mentioned, a major revision of Section 3 is imperative, but this is actually easy because we will just say (in Sec.3) that here is a variation of the important results in [Li et al. 2019]. Now we know the important reference of [Li et al. 2019] (again, our bad, and thank you), stating things in a fair and factual way is the minimum we need to do, and it will be done in our revision (we will not be stupid enough to lie on OpenReview).
>
> We hope the AC and everyone involved in this process could consider our explanations. In any case, thank you for your time and helping us improve our paper significantly.

---

### Comment · Reviewer_j2p5 · 2021-09-13
**Updated score in light of AC statement and author response**

In light of the new comments by AC and the authors, I have revised my score from 6 to 4. Although there are still novel aspects of this submission (as the authors point out), the reality is that accounting for this missed reference will require substantial rewriting, not just in Section 3, but throughout the entire paper in order to reframe the scope of the contributions as well as to provide technical comparisons with Li et al. For such a large-scale revision, it does not make sense to accept the paper to this conference without providing further feedback to the authors in the form of follow-up reviews. Therefore, I recommend that the authors carefully revise the paper and resubmit to another venue.

---

> ### Author Response · Authors · 2021-09-13
> **Follow-up response to reviewer j2p5**
>
> Thanks for your update. May we try to convince again that the needed revision is not large-scale? As our results in Sec.3 are technically *not* the identical to those in [Li et al. 2019], we just need to change some descriptions and add acknowledgments of their importance. If the concern was not being able to provide further feedback to the authors in the form of follow-up reviews, (i) what we can do is to provide a revised version, asap, before the decision deadline (we just need to be told that this could be a possibility); (ii) may we note that if it is killed and resubmitted to another venue, there will be no follow-up reviews either because reviewers will likely be different?
>
> Is there any possibility that the reviewer could reconsider a score of 4, even assuming Sec.3 had zero contribution but just based on the contribution of Sec.4?

---

### Author Response · Authors · 2021-09-18
**Paper revised; final plead**

Dear Reviewers and AC,

Thanks very much again for working on our submission. We revised the paper to address the missing of the important reference of [Li et al. 2019], and we wonder if there is any chance that you're willing to take another look and reconsider your evaluation. Going down from 7776 to 5554 was a roller coaster experience to us, but the revision really was doable. We know this is a lot to ask for, but we were not aware of this important flaw during the main discussion period of Aug 10 - Sep 2. Since it seemed to us that the five of you are not only leading experts but very fair, we are pleading for your precious time and consideration -- is this really a 5554-reject work despite of the feasibility of a revision?

The double blind revision can be found at
  https://drive.google.com/file/d/18rw6sCBVvuPu7Ne2JDUwmnOHDMvRTmAn/view

In all cases, thank you.

Best wishes,

Authors

---

### Decision · Program_Chairs · 2021-09-27

**Decision:**

Reject

**Comment:**

This is a good paper, but unfortunately authors missed an important reference [Thm 1, Li et. al., 2019], and proved a result that is almost identical to the one in that paper.

Comparing [Thm 1, Li et. al., 2019] and Thms 3.3, 3.4 and Cor 3.5 in the current paper, one notices that the results are almost identical, with very minor differences. Authors were made aware of this work, and they acknowledged missing the reference in their sincere response. Authors' main contributions rely on Thms 3.3, 3.4, and given [Thm 1, Li et. al., 2019], it decreases the novelty of this paper significantly.

All the reviewers and the AC think this paper makes important contributions, but they also think adapting the paper to this missed reference from Neurips 2019 would require a major revision of their Section 3, and invalidates some of the novelty claims made in the paper.

When the authors revise their paper and resubmit to the next conference, I would not hesitate to accept the revised version should it be assigned to me.